# How Environmental Policy Stringency, Foreign Direct Investment, and Eco-Innovation Supplement the Energy Transition: New Evidence from NICs

Anam Azam 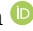

**Abstract:** Several researchers have studied the environmental policy stringency and ecological innovation regarding $CO_2$ emissions and renewable energy consumption; however, the impact of environmental policy stringency, technological innovation, FDI, and ecological innovation on energy transition has not been studied in the case of NICs. For this purpose, panel quantile regression models are applied in the context of NICs from 2000 to 2021. Our empirical results show that the effect of foreign direct investment is positive and statistically significant on energy transition. On the other hand the variables environmental policy stringency, eco-innovation, and ICT-trade have an inverse effect on energy transition. Therefore, the findings of the study also provide policy implications that indicate NICs need to optimize their trade structure and re-innovate the latest innovation spillovers, and strict environmental policies should be introduced to facilitate energy transition in NICs.

**Keywords:** energy transition; climate change; innovation; SGDs; ICT-trade

## 1. Introduction

Global warming and climate change are the main problems confronting the world due to the huge reliance on traditional energy production and consumption, which contribute more than 80% of aggregate energy consumption [1]. The over-exploitation of fossil fuel energy is due to rapidly increasing economic activities and industrialization, which significantly accelerate carbon dioxide ($CO_2$) emissions. During 2010–2019, the average global greenhouse gas emissions (GHGs) reached 54.4 gigatons ($GTCO_2e$) due to the utilization of fossil fuel energy and industrialization [2]. According to the United Nations report on climate change, the ambition is to decrease the global temperature to 1.5 °C in 2050 (UNDP, 2015) [3]; however, the continuous increase in environmental degradation issues has gained the attention of many researchers and policymakers working on energy transition.

The energy transition process includes switching from non-clean energy sources (oil, coal, and gas) to clean energy sources (wind, solar, and hydropower) as well as nuclear energy that would mitigate global $CO_2$ emissions and maintain the balance between economic growth and environmental quality. Moreover, to combat environmental degradation challenges, the United Nations introduced sustainable development goals (SDGs), whose purpose is to provide clean and sustainable energy resources, which may affect economic activities for a better quality of life [4]. Recently, developed nations have been focusing more on energy transition systems like a switch to resource and environmental energy-efficient systems because this transformation offers the development and implementation of innovative energy innovations like clean energy technologies [5]. Thus, renewable energy and nuclear power development have become important factors for a low-carbon economy. Renewable energy transition [6] is a successful new strategy that has caused strong reductions in electricity production cost and environmental pollution, as well as ensures energy security and reduces energy dependence [7]. In addition, nuclear energy promotes economic growth, and switching to nuclear energy is an effective solution [8],

reducing $CO_2$ emissions [9–11]. Thus, to understand the importance of energy transition, it is necessary to explore the determinants of energy transition.

Structural transformation in the energy sector cannot be possible without government support for environmental policies that impose environmental-related rules and regulations such as emission-reduction targets and carbon emission costs, which can promote sustainable development [6,12]. Environmental policies as well as energy structure may have different effects in developed and newly industrialized countries.

The report [13] defines the "Environmental Policy Stringency Index (EPSI) as a nationally and globally comparable measure of the environmental policy stringency" and defines the term "stringency" as "the implicit and explicit cost of ecological hazardous behavior". Moreover, the Environmental Policy Stringency Index increases renewable energy and energy efficiency through regulatory measures and strategy planning, as fiscal and financial inducements. The environmentally stringent rules and regulations aim to amend individual and organizational behavior in carbon emission ($CO_2$) mitigation by promoting consumption of less-polluting energy [14]. Most scholars have shown that environmentally stringent policy is effective in eliminating carbon dioxide emissions ($CO_2$) [3,15–17] and boosting renewable energy production [18–20], but no research has found the effect of the Environmental Policy Stringency Index on energy transition. However, the main purpose of this current research is to analyze the impact of the Environmental Policy Stringency Index, technological innovation, foreign direct investment, and eco-innovation on energy transition in newly industrialized countries.

The transition from traditional energy to renewable energy and nuclear power can be possible with the factor of foreign direct investment. FDI has a significant effect on aggregate energy consumption in host economies because it offers manufacturing skills, managerial experience, and new ideas and strategies for carbon emission reduction and energy-saving measures, which leads to the sustainable development of economies. Several previous works investigating the impact of FDI on renewable energy, for instance Refs. [21,22], show a positive impact on renewable energy use in 15 West African countries and Bangladesh. Moreover, Ref. [23] suggests that FDI increases energy consumption, which leads to higher energy demand in developing countries. In contrast, Ref. [24] claims FDI reduces renewable energy use in OECD countries. For aggregate energy, Ref. [25] finds no evidence of FDI on energy consumption in BRI countries.

Another widely discussed remedy for facilitating energy transition is technological innovation, which mitigate climate change hitches and can achieve SDGs around the world. Technological development and political and economic reforms can support the transformation of energy systems and help them be more competitive [26]. Several studies use patent application indicators as a proxy for technology innovation [27,28] while only a few studies use ICT, which is quantified by innovation in the field of energy and environment. For instance, Refs. [29–31] suggest that information and communication technology (ICT) is an imperative factor underpinning technologies that facilitate green energy innovations. On the same subject, the implementation of economic growth and environmental sustainability is significantly affected by the enlargement of ICT-trade that increases the cost-effective of green innovation, power demand, and energy efficiency by pinpointing the anomalies in the conventional energy networks [32]. In this modern era, firms started e-commerce businesses due to the deployment of ICT, which can offer new opportunities for creating a share of renewable energy in total energy consumption, enhancing energy efficiency, and creating a more decentralized supply system [33,34]. However, unlike previous studies, this study uses ICT-trade as a proxy for the effect of technological innovation on energy transition, which can facilitate green production and consumption for sustainable development in NIC countries.

Besides the development of technology innovation, eco-innovation is one of the most effective and environmentally friendly means of enhancing green development that has recently garnered the attention of many experts and economists [35,36]. Eco-innovation, also known as green technology, is a development process based on novel ideas, green

production, and consumption of goods and services that are cost-effective and emission-free. Eco-innovations can increase energy efficiency and diminish equipment losses and undue costs in the production system [37]. Several empirical studies focus on eco-innovation and its potential effect on the environment and sustainable development [38–44] and renewable energy consumption, with mixed findings. For instance, Ref. [45] found that eco-innovation increases renewable energy consumption in OECD countries.

In contrast, Refs. [46,47] used two models and concluded that using FMOLS eco-innovation and financial efficiency increases renewable energy consumption, but by using the quantile regression model, it decreases. Furthermore, Ref. [47] analyzed the impact of green finance and eco-innovation on energy efficiency from 1990 to 2020. By using the quantile regression method, the findings suggest that eco-innovation diminished the energy intensity in G7 economies. Nevertheless, these studies ignore the factor of energy transition with eco-innovation; however, further analysis is needed to analyze the effect of eco-innovation on energy transition (renewable energy and nuclear energy) for sustainable development. The author has incorporated a comprehensive summary of previous work regarding the effects of environmental stringency policy, technology innovation, and eco-innovation on energy transition in Table 1. In conclusion, the empirical research results are different due to different econometric models, different country selection, and study times. In the existing literature, several studies investigate macroeconomic variables, for instance aggregate energy consumption, $CO_2$ emission, economic growth, and trade, but no one study investigates the influence of environmental policy stringency, FDI, and eco-innovation on energy transition in NIC economies.

Newly industrialized countries (NICs) have a strong impact on the economic and energy fields globally. NICs belong to the group led by developing economic powers endeavoring to become more industrialized to be counterparts to "developed" countries (NICs include China, India, Turkey, South Africa, Brazil, Mexico, and Indonesia). NIC nations rely heavily on traditional energy sources (oil, coal, and gas) for energy consumption and have huge contributions to total $CO_2$ emissions. For instance, China and India are the top $CO_2$ emitters, followed by South Africa and Malaysia. The NICs are top energy consumers due to rapid economic growth and industrialization, which causes $CO_2$ emissions in the atmosphere. The NICs account for 38.855% of world energy consumers, 24.985% of the global GDP, 48.09% of the world population, and 20% of global trade. Moreover, NICs are also among the big $CO_2$ emitters as these economies released 48.847% of world emissions [48,49]. Therefore, the deployment of energy transition due to traditional high energy costs is in dire need of government support and is an indispensable choice for NICs because it offers excess energy supply and is the best option for eliminating $CO_2$.

**Table 1.** Summary of past studies.

| Author (s)/Time Study | Countries | Dependent Variable | Independent Variables | Models | Conclusions |
|---|---|---|---|---|---|
| [50]/1998–2013 | 137 income countries | Primary energy supply | GDP, coal reserves, oil reserves, financial capital | Fixed-effect | Financial capital supports ET |
| [51]/2000–2021 | G10 countries | Renewable energy | Technology innovation and GDP | Generalized methods of moments | Significant effect of technology innovation on renewable energy transition |
| [31]/1992–2015 | 6 South Asian states | Renewable energy transition (share of renewable energy consumption) | Inter-regional trade, foreign direct investment, GDP, oil prices, $CO_2$ emissions | Linear and non-linear regression | Regional trade amplifies the renewable energy transition |

**Table 1.** *Cont.*

| Author (s)/Time Study | Countries | Dependent Variable | Independent Variables | Models | Conclusions |
|---|---|---|---|---|---|
| [52]/1970–2014 | Lower-upper- and high-income countries | $CO_2$ emissions | Renewable energy, globalization, fossil fuel energy | ARDL | Fossil fuel consumption increases the environment, and renewable energy lessens the environment |
| [53]/1995–2015 | 38 IEA countries | GDP | Energy transition, economic sustainability, capital labor, renewable and non-renewable energy | AMG and FMOLS | Significant effect of ET on economic growth |
| [54]/1994–2019 | 26 EU countries | $CO_2$ emissions | Climate technology, energy transition, environmental regulations, GDP, urbanization | CCCE-MG | Climate technologies, energy transition, and environmental regulation diminish $CO_2$ emissions |
| [55]/1990–2015 | 16 APEC countries | GDP | Renewable energy consumption, non-renewable energy consumption, GDP, trade openness | Cup-FM | Renewable and non-renewable energy consumption increase economic growth |
| [56]/1990–2015 | 45 Asian countries | Energy transition (renewable and fossil fuel energy consumption) | GDP, exchange rate, $CO_2$ emissions, population growth | GMM | Significant effect of economic growth on energy transition |
| [18]/1984–2019 | 107 income countries | | | Regression analysis | Significant effect of renewable energy and non-renewable energy on economic growth |
| [26]/1993–2018 | Russia | Energy transition (share of renewables to non-renewables) | Inflation, $CO_2$ emissions, exchange rate, GDP, population growth, financial openness, geopolitical risk | ARDL | Eco-innovation decreases $CO_2$ emissions |
| [57]/1993–2019 | China | Environmental pollution | Environmental regulation, renewable energy consumption, GDP, FDI, environmental policy stringency | NARDL | Environmental policy stringency lessens $CO_2$ emissions |
| [58,59]/1993–2012 | Visegrad Group | Renewable energy supply | Environmental policy stringency, GDP, $CO_2$ emissions | ARDL | Environmental policy stringency increases renewable energy production |

Based on the above-mentioned gaps, the present study extends the literature in the following ways: First, this study investigates the influence of environmental policy stringency, technological innovation, foreign direct investment, and eco-innovation on energy transition (including electricity production from renewable and nuclear sources) on NICs, as most of the previous studies investigate renewable energy consumption. Second, this study investigates the effect of the Environmental Policy Stringency Index on energy transition, as most of the previous studies have investigated the relationship between environmental policy and $CO_2$ emission. Third, the author includes technological innovation as a proxy for trade to analyze the influence of ICT-trade on energy transition in the context of newly industrialized countries. The work of [31] analyzed the effects of ICT-trade on the environment in South Asia, but this study ignored the important factor of energy transition in NICs. Fourth, this study uses unique methodology of panel quantile regression and heterogeneous causality to determine the direction of causality between variables.

## 2. Materials and Methods

### 2.1. Quantile Regression Method

This study analyzes the impact of the Environmental Policy Stringency Index, technological innovation, and eco-innovation on energy transition in newly industrialized countries through the fixed-effect quantile regression model proposed by [59] to overcome the shortcomings of the traditional econometric model. Moreover, the quantile regression method is used to determine the conditional distribution in countries concerning the study variables' relationship. The advantages of this model are (1) as compared to ordinary least square regression the panel quantile regression does not require the economic sequence to be a normally distributed sequence; (2) this model is more robust against outliers in the response measurements than OLS regression; and (3) the panel quantile regression estimates the extreme values and overall influence of explanatory variables on response variables [30]. The equation of the quantile regression model is given below:

$$Qy_{it} = (\tau|X_{it}) = X'_{it}\beta_\varnothing + \varepsilon_{it} \tag{1}$$

Here, in Equation (1), $x$ indicates the vector of independent variables, $y$ is the response variables, $\varnothing$ is the quantile point, and $\varepsilon_{it}$ is the error term.

### 2.2. Theoretical Background and Model Specification

This study discusses the theoretical background before going to the econometric analysis, as it may help to determine the model's variables. Energy is produced mainly from two predominant sources, renewable and non-renewable energy. For excess electricity production, many countries rely on fossil fuel sources that emit pollution into atmosphere and harm the environment. For instance, China and India are the major consumers of fossil fuel sources, in particular coal, for electricity production.

Thus, energy transition is necessary to combat climate change and promote sustainable development. Energy transition can also be derived from several other economic factors that can help to achieve sustainable development. The first and most important factor is the Environmental Policy Stringency Index, which supports the development of energy transition. Environmental policies including public policy, feed-in tariffs, carbon taxes, and government monitoring promote energy transition [60]. According to the Porter Hypothesis, a high level of income boosts innovation and firms' ability to produce, consequently increasing energy efficiency of the economy. Technological innovation and environmentally related technologies are crucial factors for boosting the energy transition, but it requires huge green investment to improve the energy structure to facilitate the clean energy demand. Now, for developing or newly industrialized countries, capacity building for innovations in pursuit of energy transition may not be immediately possible; thus, these nations rely on technology trade, which allows cross-border low-carbon technology transfer. However, one must also remember that with more investment, energy transition can be achieved because foreign direct investment offers a huge amount of capital and green

innovation [61]. Given this, Figure 1 indicates the associated model of dependent and independent variables.

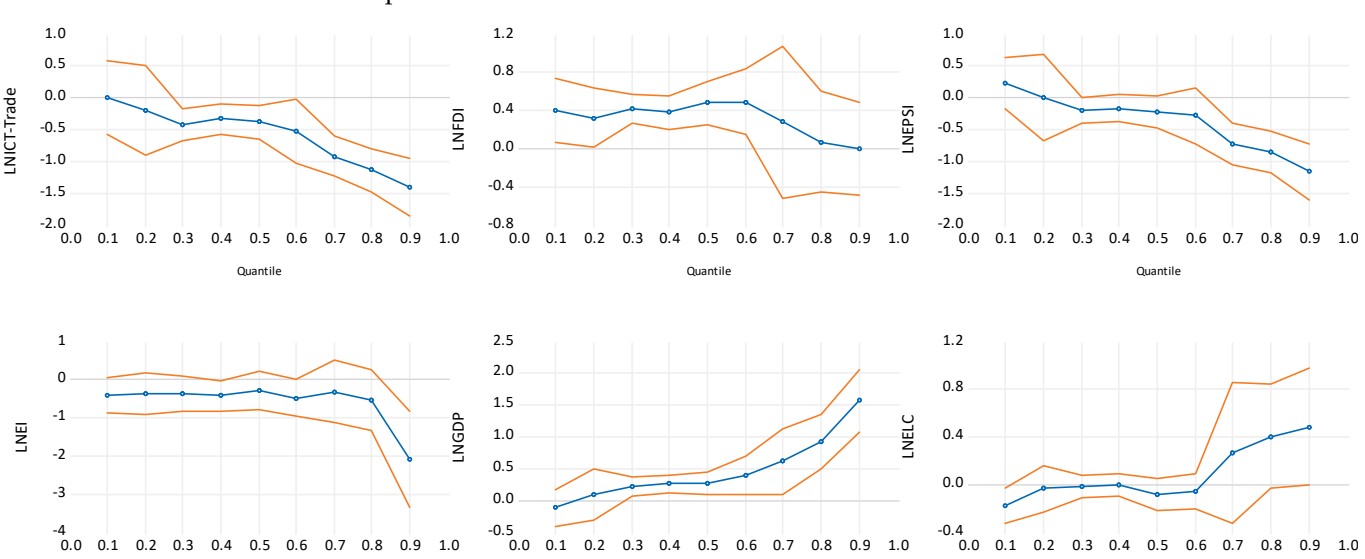

**Figure 1.** Panel quantile regression coefficients. The blue line illustrations the estimated coefficients and red line designates the 90% confidence interval.

Following the study of [26], involving an analysis for Russia, we assume the effect of independent variables on energy transition using electricity production instead of consumption in newly industrialized countries. The model is as follows:

$$ETR_{it} = f\left(EPSI_{it},\ ICTR_{it},\ ECI_{it},\ FDI_{it},\ GDP_{it},\ ELC_{it}\right) \tag{2}$$

This study transforms all the variables into natural logarithms to attain the estimated coefficients in elastic form for analysis for the panel countries. Moreover, the natural algorithm presents more reliable and efficient results [62]. Equation (2) can be transformed into natural algorithms as shown in Equation (3):

$$lnETR_{it} = \beta_0 + \beta_1\,lnEPSI_{it} + \beta_2\,lnICTR_{it} + \beta_3\,lnECI_{it} + \beta_4\,lnFDI_{it} + \beta_5\,lnGDP_{it} + \beta_6\,lnELC_{it} + \varepsilon_{it} \tag{3}$$

In the above Equations (2) and (3), ETR refers to energy transition, EPSI represents the Environmental Policy Stringency Index, ICTR denotes information and communication technology exports + imports of ICT goods and services, ECI indicates eco-innovation, FDI is foreign direct investment, GDP is economic growth, and ELC is electricity consumption. The symbol $i$ indicates 7 newly industrialized countries (Brazil, China, India, Indonesia, Mexico, Turkey, and South Africa), $t$ is a period from 2000 to 2021, $\varepsilon_{it}$ indicates the standard error, $\beta_0$ is a constant term, and $\beta_1$ to $\beta_6$ are slope coefficients of variables. The quantile regression for exploring energy transition by following Equation (1) and using the variables in Equation (3) is constructed in Equation (4):

$$Q_\tau lnETR_{it}(\tau X_{it},\ \varepsilon_i) = \beta_{0\tau} + \beta_{1\tau}lnEPSI_{it} + \beta_{2\tau}lnICTR_{it} + \beta_{3\tau}lnECI_{it} + \beta_{4\tau}lnFDI_{it} + \beta_{5\tau}lnGDP_{it} + \beta_{6\tau}lnELC_{it} + \varepsilon_{it} \tag{4}$$

In Equation (4), $\beta_{0\tau}$ is the constant term, $\tau$ is a quantile measure, $\beta_{1\tau}$ to $\beta_{6\tau}$ are the estimation parameters, and $Q_\tau lnETR_{it}(\tau X_{it},\ \varepsilon_i)$ denotes the quantile function.

### 2.3. Data and Variable Description

The purpose of the current study is to analyze the impact of the Environmental Policy Stringency Index, technological innovation, and eco-innovation on energy transition in newly industrialized countries from 2000 to 2021. Unlike previous studies, the dependent variable of this study is the energy transition share from fossil fuel sources to both

renewables and nuclear energy. The independent variables are the Environmental Policy Stringency Index (EPSI), which ranges from 0 to 6 (not stringent and highly stringent); ICT-trade openness measured as a sum of exports and imports of ICT goods and services as a percentage of GDP; eco-innovation measured as number of parents; foreign direct investment (net inflows measured as a percentage of GDP), where GDP is measured in current USD; and ELC measured as total electricity consumption. The data for energy transition are obtained from [63]. The data on environmental policy stringency and eco-innovation are obtained from the OECD source, while data on ICT-trade, foreign direct investment, and GDP are collected from the World Bank. Variable descriptions, sources, and units are presented in Table 2.

**Table 2.** Description of variables and data sources.

| Variables | Description and Abbreviations | Units | Sources |
|---|---|---|---|
| Energy transition | Share of renewables and nuclear energy generation to fossil fuel sources (ETR) | % | Energy institute statistical review of world energy |
| Environmental policy stringency | Environmental Policy Stringency Index (EPSI) | - | OECD statistics |
| ICT-trade | Information and communication technology exports + imports of ICT goods and services (ICTR)s | (% of GDP) | World Bank database |
| Eco-innovation | Patents on environmental-related technologies (ECI) | Number | OECD statistics |
| Foreign direct investment | Foreign direct investment, net inflows (FDI) | (% of GDP) | World Bank database |
| GDP | Economic growth | Current USD | World Bank database |
| ELC | Electricity consumption | | |

Notes: OECD indicates the Organization for Economic Co-operation and Development; GDP indicates economic growth.

## 3. Results and Discussion

### 3.1. Summary Statistics

Table 3 highlights the results of descriptive statistics of all variables. The factors ICT-trade, eco-innovation, and GDP have the highest mean values. The results show that the distribution of all variables is skewness. The energy transition and foreign direct investment are leptokurtosis, and other variables are platykurtosis, which means they have few outliers. This study employs the Shapiro–Wilk normality test before the estimation of the panel quantile regression model. Table 4 indicates the findings of the normality test; based on the *p*-values the results show a non-normal distribution of the data. The findings of the panel quantile regression are more effective when the data have a non-normal distribution.

**Table 3.** Descriptive statistics.

| Variables | Mean | Std.Dev | Min | Max | Skewness | Kurtosis | Jarque–Bera Test |
|---|---|---|---|---|---|---|---|
| ET | 1.257 | 2.654 | 0.059 | 11.628 | 2.541 | 8.172 | 337.48 [a] |
| ICT-trade | 9.465 | 7.739 | 0.000 | 28.394 | 1.121 | 2.736 | 32.748 [a] |
| FDI | 4.500 | 6.640 | −4.550 | 3.440 | 2.525 | 8.980 | 393.168 [a] |
| EPSI | 1.142 | 0.858 | 0.170 | 3.140 | 0.831 | 2.536 | 19.128 [a] |
| EI | 9.351 | 2.945 | 2.320 | 19.320 | 0.329 | 3.172 | 2.971 [a] |
| GDP | 8.430 | 0.849 | 6.091 | 9.488 | −0.920 | 2.913 | 21.225 [a] |
| ELC | 3.368 | 1.235 | 1.258 | 6.479 | 0.638 | 3.100 | 10.269 |

Notes: [a] indicates significance at 1%.

**Table 4.** Shapiro–Wilk test.

| Variables | Test Statistics (*p*-Value) |
|---|---|
| ET | 0.779 [a] (0.000) |
| ICT-trade | 0.861 [a] (0.000) |
| FDI | 0.978 [a] (0.019) |
| EPSI | 0.936 [a] (0.000) |
| EI | 0.991 (0.468) |
| GDP | 0.939 (0.000) |
| ELC | 0.531 (0.000) |

Notes: [a] indicates significance at 1%.

### 3.2. Results of Pearson Correlation Test

The Pearson correlation test is used to determine the existence of a positive or negative relationship between a dependent variable and an independent variable. The results are illustrated in Table 5.

**Table 5.** Results of Pearson correlation.

| Dependent Variable | Independent Variables | | | | | |
|---|---|---|---|---|---|---|
| | EPSI | FDI | ICT-trade | EI | GDP | ELC |
| ET | −0.216 [a] | 0.374 [a] | −0.268 [a] | −0.038 | 0.276 [a] | 0.277 [a] |

Notes: [a] indicates significance at 1%.

The findings in Table 5 represent that there is a negative relationship between environmental policy stringency, ICT-trade, and eco-innovation, whereas for FDI, GDP, and ELC it indicates a positive correlation with energy transition in newly industrialized countries.

### 3.3. Slope Heterogeneity Test

Another heterogeneity test is used to check whether the slope of coefficients is homogeneous or heterogeneous [64]. Table 6 represents the results of the slope heterogeneity test developed by [65]. According to the findings, the coefficients are not homogenous, and *p* values are significant at the 1% level. As a result, the null hypothesis of the slope coefficient is rejected, and the alternative hypothesis of slope heterogeneity is used.

**Table 6.** Results of heterogeneity test.

| | Delta | *p*-Value |
|---|---|---|
| Delta | −2.375 [a] | 0.000 |
| Delta$_{adj}$ | −2.977 [a] | 0.000 |

Notes: [a] indicates significance at 1%.

### 3.4. Cross-Sectional Dependence Test

In panel data econometrics, the next step is the cross-sectional dependence test. The results of the cross-sectional dependence (CSD) model proposed by [66] are in Table 7. The findings indicate that all factor's *p*-values are highly significant, which means that all variables such as ET, ICT-trade, FDI, EPSI, EI, GDP, and ELC have the existence of CSD.

**Table 7.** Cross-sectional dependence analysis.

| Variables | Test Statistics (*p*-Value) |
|---|---|
| ET | 14.10 [a] (0.000) |
| ICT-trade | 16.80 [a] (0.000) |
| FDI | 17.17 [a] (0.000) |
| EPSI | 21.18 [a] (0.000) |
| EI | 5.368 [a] (0.000) |
| GDP | 8.759 [a] (0.000) |
| ELC | 3.029 [a] (0.000) |

Notes: [a] indicates significance at 1%.

### 3.5. Results of Panel Unit Root Test and Panel Co-Integration Test

This study used the Augmented Dickey–Fuller (ADF) [67] and Philips–Perron unit root [61] tests to identify whether the variables are stationary because the empirical econometric analysis series must be stationary to avoid simulated findings. Table 8 shows the findings of ADF and Philips–Perron unit root tests for this intent. The results indicate that most of the variables' energy transition, foreign direct investment, Environmental Policy Stringency Index, GDP, and ELC are non-stationary at a level except ICT-trade and eco-innovation, and they turned stationary at the first difference in the ADF unit root test but non-stationary in the Fisher–PP unit root test. Therefore, it can be concluded that the variables are in the first-order integer, and we can continue to perform the regression model on the panel data.

**Table 8.** Results of unit root test.

| Variables | Fisher–ADF | | Fisher–PP | |
|---|---|---|---|---|
| | Level | First Diff. | Level | First Diff. |
| ET | 11.85 | 50.17 [a] | 15.038 | 124.96 [a] |
| ICT-trade | 25.43 [a] | 67.84 [a] | 20.615 | 119.40 [a] |
| FDI | 11.98 | 68.54 [a] | 35.399 [a] | 373.72 [a] |
| EPSI | 8.85 | 37.86 [a] | 9.128 | 78.424 [a] |
| EI | 33.9 [a] | 110.4 [a] | 57.768 [a] | 714.05 [a] |
| GDP | 6.863 | 38.88 [a] | 13.38 | 62.37 [a] |
| ELC | 10.68 | 48.91 [a] | 20.77 | 100.13 [a] |

Note: [a] represents 1% significance level.

Next, the panel co-integration test is required to judge whether the variables are co-integrated in the long run. For this, this study used the Johansen Fisher panel co-integration test that is proposed by [68]. Table 9 indicates that the findings accept the alternative hypothesis and reject the null hypothesis of no co-integration, which means that variables are co-integrated and have a long-run relationship.

**Table 9.** Results of Johansen Fisher panel co-integration test.

| Hypothesized No. of CE (s) | Fisher Stat. * (From Trace Test) | Prob. | Fisher Stat. * (From Max-Eigen Test) | Prob. |
|---|---|---|---|---|
| None | 153.7 [a] | 0.000 | 114.1 [a] | 0.000 |
| At most 1 | 63.94 [a] | 0.000 | 52.47 [a] | 0.000 |
| At most 2 | 27.35 [a] | 0.017 | 22.27 [b] | 0.073 |
| At most 3 | 93.75 [a] | 0.076 | 47.01 [a] | 0.076 |
| At most 4 | 14.95 | 0.381 | 10.79 | 0.7020 |
| At most 5 | 29.99 [a] | 0.002 | 23.60 [b] | 0.023 |
| At most 6 | 24.67 [b] | 0.0165 | 24.67 [b] | 0.0165 |

Notes: [a,b] indicate significance at 1%, 5%, respectively.

### 3.6. Results of Panel Quantile Regression

This study used a panel quantile regression model because the data sets are non-normal distributions. Here, we divide the outcomes into (0.1, 0.2, . . ., 0.9) quantiles. The results of the regression model are shown in Table 10 and Figure 1; as each quantile can adequately describe the distribution characteristics of energy transition, the quantile regression model directly indicates the effects of independent variables on dependent variables. The results show that the coefficient of ICT-trade is negative and insignificant in the lower quantiles, but it is inverse and statistically significant in the middle and higher quantiles.

**Table 10.** Results of panel quantile regression.

| Variables | Quantile Statistics | | | | | | | | |
|---|---|---|---|---|---|---|---|---|---|
| | Lower Quantile | | | Middle Quantile | | | Higher Quantile | | |
| | 0.1 | 0.2 | 0.3 | 0.4 | 0.5 | 0.6 | 0.7 | 0.8 | 0.9 |
| ICT-trade | 0.010 | −0.046 | −0.334 [a] | −0.322 [a] | −0.344 [a] | −0.342 [b] | −0.921 [a] | −1.112 [a] | −1.485 [a] |
| FDI | 0.391 [b] | 0.321 [a] | 0.382 [a] | 0.343 [a] | 0.390 [a] | 0.149 | 0.149 | −0.030 | ⁻0.014 |
| EPSI | 0.228 | −0.031 | −0.157 | −0.156 | −0.188 | −0.310 | −0.801 [a] | −0.884 [a] | −1.206 [a] |
| EI | −0.402 [c] | −0.29 | −0.298 | −0.431 [b] | −0.235 | −0.163 | −0.294 | −0.467 | −1.733 [a] |
| GDP | −0.119 | 0.085 | 0.172 [b] | 0.253 [a] | 0.262 [a] | 0.349 [a] | 0.675 | 0.933 [a] | 1.370 [a] |
| ELC | −0.176 [b] | −0.036 | −0.015 | 0.003 | −0.026 | 0.128 | 0.360 | 0.460 [a] | 0.413 [b] |

Notes: [a, b, c] represent 1%, 5%, and 10% significance levels, respectively.

The negative sign indicates that a 1% increase in ICT-trade is accompanied by a reduction in energy transition, which is consistent with [31]. This outcome implies that these countries have higher levels of non-renewable energy consumption than renewable energy consumption due to the high implications. Environmental and health gains are associated with using green energy, but due to less bilateral trade among countries decreasing use of green energy by NIC countries, a small share of ICT goods and services may not be enough to provoke the energy transition phenomenon in NICs.

Apart from the coefficient of ICT-trade, foreign direct investment is positive and significant at (0.1–0.5) quantiles regarding energy transition, but it is negative and insignificant at higher quantiles, which means a 1% increase in net inflows of foreign direct investment will improve energy transition in NIC countries. This outcome is similar to the study of [69], who concluded that FDI inflows have a considerable significant impact on energy consumption. This implies that an increase in FDI inflows has a significant effect on energy transition implying that that NICs might have focused on energy transition (from fossil fuel to renewable and nuclear energy) for two primary reasons: first, to meet the increasing energy demand and second, to eliminate the $CO_2$ emissions as energy transition is free from emissions.

The influence of environmental policy stringency on energy transition is negative and insignificant at lower (i.e., 0.2–0.6) quantiles, implying that it does not affect energy transition, but it is negative and significant at higher (i.e., 0.7–0.9) quantiles, which indicates that environmental policies encumber energy transition in these selected countries. This outcome is in line with [12,20], in which the authors conclude that environmental policy stringency has a detrimental effect on renewable energy consumption for selected BRICST countries. Moreover, according to our results, environmental policy stringency is laxer and not fully backed by law in NICs; as a result, countries use non-renewable energy. Similarly, negative effects on energy transition are ascertained in the context of eco-innovation. These findings are in line with [41,46]. However, the negative impact of environmental policy stringency and eco-innovation is based on the fact that innovation related to the environment may not be directed toward energy transition in these countries, and they still rely on fossil fuel energy sources. Moreover, a surge in energy transition is caused by a surge in GDP at each quantile, implying that an increase in economic growth explains an increase in energy transition at middle and higher quantiles. Finally, electricity consumption contributes to an increase in energy transition at higher quantiles, which indicates that electricity demand is the major driver of energy transition in NIC economies.

Finally, this study uses the panel causality test developed by [70] to test whether the variables have a causal relationship or not. Table 11 shows the findings of the panel causality test; here, the results are significant at 10% and 5% levels, respectively. There is unidirectional causality from ICT-trade, foreign direct investment, and other variables to energy transition; there is no evidence of causality from eco-innovation to energy transition.

**Table 11.** Results of panel quantile regression. Results of the panel DH-causality test.

| Null Hypothesis (H$_0$) | W-Stat. | *p*-Value | Decision |
|---|---|---|---|
| ICT-trade $\nRightarrow$ ET | 3.991 | 0.098 [c] | Unidirectional causality |
| ET $\nRightarrow$ ICT-trade | 1.131 | 0.248 | |
| FDI $\nRightarrow$ ET | 4.087 | 0.080 [c] | Unidirectional causality |
| ET $\nRightarrow$ FDI | 2.777 | 0.644 | |
| EPSI $\nRightarrow$ ET | 4.451 | 0.035 [b] | Unidirectional causality |
| ET $\nRightarrow$ EPSI | 2.959 | 0.522 | No causality |
| EI $\nRightarrow$ ET | 3.244 | 0.357 | |
| ET $\nRightarrow$ EI | 2.955 | 0.544 | |
| GDP $\nRightarrow$ ET | 6.293 | $9 \times 10^{-5}$ [a] | Unidirectional causality |
| Et $\nRightarrow$ GDP | 1.803 | 0.620 | |
| ELC $\nRightarrow$ ET | 4.665 | 0.020 [b] | Unidirectional causality |
| ET $\nRightarrow$ ELC | 3.543 | 0.224 | |

Notes: [a, b, c] indicate significance at 1%, 5%, and 10%.

## 4. Conclusions and Policy Implications

This paper investigates the effect of environmental policy stringency, technological innovation as a proxy for ICT-trade, eco-innovation, and foreign direct investment on energy transition (generation of renewable energy and nuclear power rather than reliance on fossil fuels) from 2000 to 2021 in the context of newly industrialized countries. For this, we employed different statistical methods, such as the cross-sectional dependence test, the unit root test to check the integration order, and the co-integration test, and panel quantile regression in different quantiles was used to analyze the asymmetries long-run results. We also employed the panel D-H causality test to determine the causality between series. The quantile regression results show that environmental policy stringency, eco-innovation, and ICT-trade have an inverse effect on energy transition, while foreign direct investment, GDP, and electricity consumption significantly influence energy transition. There is one-way causality between ICT-trade, FDI, EPS, ELC, and GDP and energy transition, but there is no causality found between eco-innovation and energy transition. Thus, our empirical results present some important suggestions for NIC policymakers.

According to our results, energy transition is affected by many factors. The negative effect of environmental policy stringency indicates that NIC countries have an experience of less immature technology progress in the energy sector. Therefore, governments need to introduce environmental policies in the energy transition to transform the NICs into energy-efficient countries. For this, they should support different projects with strategies such as subsidies and finance access for energy transition.

Another critical factor that affects energy transition is foreign direct investment. Our results have shown that FDI is the major contributor to the development of the energy transition process in the short term and long term. This means that an increase in foreign direct investment could improve the speed of progress of energy transition in newly industrialized countries. Foreign direct investment increases energy transition through the improvement of economic activities in NICs. Therefore, attracting more FDI in newly industrialized countries will increase investment in the market, which will encourage the use of innovations and technologies that are more energy efficient, which leads to use of more clean energy. The government should prioritize channeling FDI for the development of the clean energy sector, which can increase the energy security in NICs.

Via regression analysis, we have also found that ICT-trade reduces energy transition. ICT goods increase the electricity demand through digital technologies and make energy more efficient. Therefore, NICs should focus on the adoption of trade liberalization policies because a reduction of trade barriers can increase green ICT goods and services that are traded and have the capacity to employ green energy sources, so that they can implement energy transition. Moreover, NICs need to focus on the adoption of environment-friendly innovations by offering lower interest rates on purchases of energy transformation innovations such as solar panels and electric vehicles, which will make major strides in energy

efficiency that will facilitate boosting energy transition in NICs. NICs' economic growth has often led to an increase in energy transition; thus, policy makers need to cautiously implement economic development policies that aim to promote energy transition (from fossil fuel to more clean energy) by improving the green economic system and green products and developing green technologies and innovations. NICs are entering a period of economic development to achieve their objectives of global industrialization for efficient infrastructure improvement. The energy industry, through innovation, cost reduction, and collaboration, will continuously promote more economic activities within regions, create more investment in the host country, and promote a higher growth rate. Energy transition is tightly linked to economic development, but a higher level of economic growth corelates with hugely reliable and efficient electricity demand. Therefore, for high rates of electricity consumption by end-users, governments should produce a share of total electricity from clean energy sources to meet the energy demand in NICs.

Like with other studies, this study has some limitations. First, this study is for newly industrialized countries; further research can be conducted for developed, developing, and other groups of countries to compare energy transition. Second, future studies can achieve significant results by incorporating new variables such as interest rates, R&D using patent data, political factors such as good governance, etc. Moreover, other econometric models can be included to carry out future studies.

**Funding:** This research received no external funding.

**Institutional Review Board Statement:** Not applicable.

**Informed Consent Statement:** Informed consent was obtained from all subjects involved in the study.

**Data Availability Statement:** Data is available on request.

**Conflicts of Interest:** The authors declare no conflict of interest.

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
