# Peer review of "How Environmental Policy Stringency, Foreign Direct Investment, and Eco-Innovation Supplement the Energy Transition: New Evidence from NICs"

_sustainability, doi:10.3390/su16073033_

Round 1

Reviewer 1 Report

Comments and Suggestions for Authors

Report Review on the paper

How environmental policy stringency, technological innovation and ecological innovation supplement the energy transition in the SDGs framework: New evidence from Newly Industrialized Countries"

This paper investigates the impact of environmental policy stringency, technological innovation, FDI, and ecological innovation on energy transition in the case of NICs.

However, I believe the authors should improve their work by considering the following suggestions.

1.      In Results of Panel Quantile Regression section, the authors present the results of panel quantile regression for eight (0.1, 0.2,......0.8) quantiles.  Why do the authors ignore the 0.9 quantile?  Policy makers are generally more interested in the events that are the tail of the distributions, particularly lower and upper tails. The analysis should include the 0.9 quantile.

2.      In section 3.1 (Summary statistic), lines 221 and 222. It seems that this sentence is strange “However, the estimated coefficient of the quantile regression model is more robust for empirical analysis”. It makes no sense in this context. Moreover, the justification of the suitability of the panel quantile regression model is very restricted.  Before implementing quantile regression, it should be ensured that the variables under study are not normally distributed. The authors should check the normality of the data by using more tests (Kolmogorov-Smirnov test, the Shapiro-Wilk test, and the Anderson Darling test) and plot the normal Q-Q plot.

3.      In Table 4, the results of Pearson correlation are presented. However, no test was performed to check for the statistical significance of the coefficient of correlation.

4.      In the panel framework, it is important to conduct the Slope homogeneity test. The Pesaran and Yamagata (2008) slope homogeneity test checks whether the slopes of the independent variables are homogeneous across different individuals in a panel dataset. It is valuable when estimating a fixed effects panel model. See for instance, https://doi.org/10.1016/j.bir.2023.10.001

5.      The paper has to be edited for English language issues before it is ready for publication.

Overall, the study requires a comprehensive revision to cross the barrier to be published in such an esteemed journal.

Comments on the Quality of English Language

Average

Author Response

Reviewer 1. This paper investigates the impact of environmental policy stringency, technological innovation, FDI, and ecological innovation on energy transition in the case of NICs.

However, I believe the authors should improve their work by considering the following suggestions.

  1. In Results of Panel Quantile Regression section, the authors present the results of panel quantile regression for eight (0.1, 0.2,......0.8) quantiles.  Why do the authors ignore the 0.9 quantile?  Policy makers are generally more interested in the events that are the tail of the distributions, particularly lower and upper tails. The analysis should include the 0.9 quantile.

Response: Thanks for your suggestions.  In Results of Panel Quantile Regression section, I have explained 0.9 quantile and highlighted in table.  

  1. In section 3.1 (Summary statistic), lines 221 and 222. It seems that this sentence is strange “However, the estimated coefficient of the quantile regression model is more robust for empirical analysis”. It makes no sense in this context. Moreover, the justification of the suitability of the panel quantile regression model is very restricted.  Before implementing quantile regression, it should be ensured that the variables under study are not normally distributed. The authors should check the normality of the data by using more tests (Kolmogorov-Smirnov test, the Shapiro-Wilk test, and the Anderson Darling test) and plot the normal Q-Q plot.

Response: In the (Summary statistic), lines 221 and 222 have been corrected. Furthermore, the author has applied Shapiro-Wilk test to check the normality. 

  1. In Table 4, the results of Pearson correlation are presented. However, no test was performed to check for the statistical significance of the coefficient of correlation.

Response: Yes, in the section of Pearson correlation test the statistical significance of the coefficient of correlation is performed in the manuscript.

.

  1. In the panel framework, it is important to conduct the Slope homogeneity test. The Pesaran and Yamagata (2008) slope homogeneity test checks whether the slopes of the independent variables are homogeneous across different individuals in a panel dataset. It is valuable when estimating a fixed effects panel model. See for instance, https://doi.org/10.1016/j.bir.2023.10.001

Response: Thanks for suggesting this test for research's homogeneity test. Yes, the author has apply Pesaran and Yamagata (2008) slope homogeneity test checks.

  1. The paper has to be edited for English language issues before it is ready for publication.

Overall, the study requires a comprehensive revision to cross the barrier to be published in such an esteemed journal.

Response: Yes, I have checked the whole manuscript for English language and correct it.

Reviewer 2 Report

Comments and Suggestions for Authors

This paper mainly studies the impact of environmental policy stringency, technological innovation, FDI and ecological innovation on energy transition of Newly Industrialized Countries. The arithmetic model was established by the quantile regression method, and the conclusions were obtained by the Jarque-Bera test, the cross-sectional test and the co-integration test, and its reliability was verified. The results show that FDI has a significant positive impact on energy transition, and environmental policy stringency, ecological innovation and ICT trade have a negative impact on energy transition. Finally, corresponding policy recommendations are provided. The study contributes to the energy transition and sustainable development of Newly Industrialized Countries.

However, there are still the following suggestions and doubts.

1. The topic should also highlight the prominent factor of foreign direct investment.

2. The format of the paper could be further improved.

3. The conclusion of the summary is contradictory, and the conclusion is that the stringency of environmental policies has a negative impact on the energy transition, so why is it recommended to establish strict policies?

4. Does the second paragraph of section 3.2.2 begin with a shift from fossil fuels to renewable energy and nuclear energy?

Comments on the Quality of English Language

Minor editing of English language required

Author Response

Reviewer 2.

This paper mainly studies the impact of environmental policy stringency, technological innovation, FDI and ecological innovation on energy transition of Newly Industrialized Countries. The arithmetic model was established by the quantile regression method, and the conclusions were obtained by the Jarque-Bera test, the cross-sectional test and the co-integration test, and its reliability was verified. The results show that FDI has a significant positive impact on energy transition, and environmental policy stringency, ecological innovation and ICT trade have a negative impact on energy transition. Finally, corresponding policy recommendations are provided. The study contributes to the energy transition and sustainable development of Newly Industrialized Countries.

However, there are still the following suggestions and doubts.

  1. The topic should also highlight the prominent factor of foreign direct investment.

Response: Thanks for your suggestions. The title of the manuscript has been corrected and add the foreign direct investment in the topic that is highlighted.

  1. The format of the paper could be further improved.

Response: The format of the paper has been improved.

  1. The conclusion of the summary is contradictory, and the conclusion is that the stringency of environmental policies has a negative impact on the energy transition, so why is it recommended to establish strict policies?

Response: The conclusion section have been corrected with the influence of environmental policy stringency.

  1. Does the second paragraph of section 3.2.2 begin with a shift from fossil fuels to renewable energy and nuclear energy?

Response: Section 3.2. is the methodology test for correlation in the manuscript.

Reviewer 3 Report

Comments and Suggestions for Authors

Please read the attachment. Thank you.

Author Response

  • Title: the title is too Please shorten it to no more than ten words.

Response: Yes, I have short the topic as suggested!

  • Keywords: Please provide between 5 and 10 keywords that do not repeat the words or phrases in the title.

Response: Regarding the keywords, Yes, I have correct it in the manuscript. 

  • Introduction: please add a section to introduce the manuscript

Response: Yes, i have added as suggested.

  • Equations should be aligned

Response: The equations have been corrected.

  • The main text should mention or explain all equations, figures, and

        Response: All the equations with figures and tables are mentioned in the manuscript.

  • Table 8 should be adjusted and

Response: Table 8 has been corrected.

  • There is no subsection 2. Please check it.

Response: Yes, i have added in the manuscript.

  • Literature review: There is a limited literature review on the work related to the assessment method for renewable and nuclear energies and other new The following work could be helpful. (i) Renewal Energy Efficiency Assessment.

Response: Yes i have added literature review on the work related to the assessment method for renewable and nuclear energies and other new approaches.

  • The manuscript addresses an important and relatively underexplored area by examining the impact of environmental policy stringency, technological innovation, FDI, and ecological innovation on energy transition, specifically within the context of NICs. This contributes to filling a gap in the literature and offers valuable insights into the dynamics of energy transition in rapidly developing

Response: Thanks for the comments.

  • How might the   inclusion   of   additional   control   variables,   such   as

institutional factors or government policies specifically targeting energy transition, potentially enhance the robustness of the empirical analysis conducted in the manuscript?

         Response: Thanks for the suggestion to include other additional factors for      robustness of the empirical analysis. Yes, I have added the additional variables in the manuscript.

  • Could the authors elaborate on potential mechanisms underlying the observed inverse relationship between environmental policy stringency,

eco-innovation, ICT-trade, and energy transition in NICs? Are there any specific contextual factors or industry dynamics that may help explain these findings in greater detail?

  • Response: Yes, i have explained inverse relationship between environmental policy stringency, eco-innovation, ICT-trade, and energy transition in NICs in the conclusion section and empirical results section in detail. Other variables like can help also to explain according to the findings.
  • References should be formatted following the journal

Response: References have been correct following the journal template.

The reviewer hopes his point of view could help the author improve their work. Thank you for reading.

Response: Thanks for your suggestion and nice comments, these suggestions help the author to improve his work. 

Round 2

Reviewer 1 Report

Comments and Suggestions for Authors

Dear authors,

I appreciate the efforts made by the authors in order to improve the paper. However, I noted that the variables of the model have been changed due to the introduction of 2 new variables (GDP and ELC) and therefore the results have been modified compared to the initial version.

The authors should explain the motivation behind these changes that affected the results presented in the initial version.

Author Response

Author response to reviewer

I appreciate the efforts made by the authors in order to improve the paper. However, I noted that the variables of the model have been changed due to the introduction of 2 new variables (GDP and ELC) and therefore the results have been modified compared to the initial version.

The authors should explain the motivation behind these changes that affected the results presented in the initial version.

Author's response: Thank you very much for your time and kind response regarding my submission. Actually, I added two additional variables in an attempt to improve the robustness of the empirical analysis. This has enabled me to elaborate my findings in a better and clear way.   
